# *Lef1* and *Dlx3* May Facilitate the Maturation of Secondary Hair Follicles in the Skin of Gansu Alpine Merino

**DOI:** 10.3390/genes13081326

**Published:** 2022-07-25

**Authors:** Hongxian Sun, Zhaohua He, Qiming Xi, Fangfang Zhao, Jiang Hu, Jiqing Wang, Xiu Liu, Zhidong Zhao, Mingna Li, Yuzhu Luo, Shaobin Li

**Affiliations:** Gansu Key Laboratory of Herbivorous Animal Biotechnology, Faculty of Animal Science and Technology, International Wool Research Institute, Gansu Agricultural University, Lanzhou 730070, China; sunhx@st.gsau.edu.cn (H.S.); hezh@st.gsau.edu.cn (Z.H.); xiqm@st.gsau.edu.cn (Q.X.); zhaofangfang@gsau.edu.cn (F.Z.); huj@gsau.edu.cn (J.H.); wangjq@gsau.edu.cn (J.W.); liuxiu@gsau.edu.cn (X.L.); zhaozd@gsau.edu.cn (Z.Z.); limn@gsau.edu.cn (M.L.); luoyz@gsau.edu.cn (Y.L.)

**Keywords:** Gansu Alpine Merino, hair follicle, *Lef1*, *Dlx3*, spatiotemporal expression

## Abstract

Lymphatic enhancer factor 1 (*Lef1*) and distal-less homeobox 3 (*Dlx3*) are the transcription factors involved in regulating hair follicle development in mice, goats, and other animals. Their deletion can lead to hair follicle deficiency. In this study, hematoxylin–eosin staining (HE), real-time quantitative PCR (RT-qPCR), immunohistochemistry, and immunofluorescence were used to analyze the expression, location, and biological functions of *Lef1* and *Dlx3* in the lateral skin of Gansu Alpine Merino aged 1, 30, 60, and 90 days. The results revealed that the number of hair follicles decreased with age and was significantly higher at 1 day than in the other three age groups (*p* < 0.05). The mRNA levels of *Lef1* and *Dlx3* in the skin of 30-day old Gansu Alpine Merino were significantly higher than those in the other three age groups (*p* < 0.05). Protein expression of Lef1 and Dlx3 was lowest at 1 day (*p* < 0.05) and peaked at 60 days. *Lef1* and *Dlx3* exhibited a high density and strong positive expression in the dermal papillae; additionally, *Dlx3* exhibited a high density and strong positive expression in the inner and outer root sheaths. Collectively, *Lef1* and *Dlx3* may facilitate the maturation of secondary hair follicles, which is mainly achieved through the dermal papillae and inner and outer root sheaths.

## 1. Introduction

The growth and development of hair follicles is a complex and long-term physiological process that is regulated by various physical factors and signal pathways. Gene expression is one of the decisive factors affecting sheep wool growth [1,2]. The Wnt/β-catenin signaling pathway considerably influences the embryogenesis of the hair follicle and hair growth [3,4]. Lymphatic enhancer factor 1 (*Lef1*), a transcription factor, positively regulates the Wnt/β-catenin signaling pathway [5]. It is one of the main downstream members of this pathway, belonging to the HMG family, and is a lytic enhancer widely distributed in the feather follicle keratin promoter [6,7]. It is involved in cell proliferation, differentiation, and apoptosis [8,9]. In contrast, in a range of tissues of epithelial origin, such as hair follicles, mammary glands, and teeth, *Lef1* promotes normal development by regulating the interaction between epithelial and mesenchymal cells [7,10]. In addition, it has been suggested that *Lef1* binds to other transcription factors to form complexes that activate target transcription genes and regulate cell differentiation and self-renewal of tissues, which are also closely related to hair follicle development and self-renewal [11].

The distal-less homeobox 3 (*Dlx3*) gene exhibits similar functions to those of *Lef1*. It is located downstream of *Lef1* and exerts regulatory effects in hair follicle development by increasing the proliferation of primary and secondary dermal papilla cells and inhibiting apoptosis [12]. *Dlx3* is a member of a family of homozygous heterotypic frame transcription factors [13], and the transcription genes that are aggregated together are *Dlx1*–*Dlx2*, *Dlx3*–*Dlx4*, and *Dlx5*–*Dlx6*. Various homologous structural domain transcription factors play an important role in the growth, development, and differentiation of tissues and organs of various animals [14,15]. A study reported that mutations in the coding region of *Dlx3* lead to the development of hair-dentition-bone syndrome [16], which was characterized by traits such as tangled curly hair. Wool curl is one of the important indicators of the economic status of wool; therefore, the study of *Dlx3* has a great economic value [17,18]. Functional studies have reported that the knockdown of *Lef1* and *Dlx3* in mice results in the complete disappearance of antennal hair follicles; *Lef1* knockdown lead to cessation of dorsal skin follicle growth [19], and *Dlx3* itself knockdown affected the formation of the hair shaft and inner root sheaths within the hair follicles of mice, which later leads to embryonic lethality or very severe alopecia after birth [20]. Overexpression of *Lef1* in mice exhibited the same effect on normal growth of tentacles and hairs [21], whereas reduced expression of *Dlx3* had an abnormal effect on the growth of root sheaths within the hair follicles of mice. Keratin expression was similarly affected by *Lef1* and *Dlx3* [20]. 

Thus, *Lef1* and *Dlx3* play important regulatory roles in the development of hair follicles and hair growth. Their mutations can affect the development of hair follicles [22,23]. At present, most studies on *Lef1* and *Dlx3* have been reported in humans [16], mice [20,21], and other research subjects during the embryonic stage, and fewer studies are available on various growth and developmental stages after birth. Studies on hair follicle development in Gansu Alpine Merino in various growth stages after birth are not available. To elucidate the role of *Lef1* and *Dlx3* in postnatal hair follicle development in Gansu Alpine Merino, hematoxylin–eosin staining (HE), real-time quantitative PCR (RT-qPCR), immunohistochemistry, and immunofluorescence were used to observe the skin tissue structure of Gansu Alpine Merino at various ages. The expression of *Lef1* and *Dlx3* in the skin and localization of hair follicles were analyzed at several time periods. This study laid the foundation for analyzing the mechanism of development and maturation of hair follicles in postnatal Gansu Alpine Merino and provided some insights into developing strategies to treat hair loss in humans.

## 2. Material and Methods

### 2.1. Collection of Animal Samples

We selected three newly born ewes of Gansu Alpine Merino in Tianzhu Tibetan Autonomous County, Gansu Province, with similar body conditions and the same nutrition and environmental conditions. At 1, 30, 60, and 90 days, tissue from a 5 cm^2^ area of their lateral skin at the posterior border of the scapula was extracted using a skin sampler with a 0.88 mm diameter. The collected samples were washed using diethylpyrocarbonate-treated water. A portion of the samples was placed in lyophilization tubes with an RNA protection solution for temporary storage in liquid nitrogen tanks and brought back to the laboratory for subsequent RT-qPCR analysis. The other part was placed in lyophilized tubes containing 4% paraformaldehyde and then brought back to the laboratory in a 4 °C ice box for subsequent HE, immunohistochemistry, and immunofluorescence studies. Each sampling was performed after anesthesia using 2% lidocaine injection at the sampling site, and penicillin (Jiangxi Ruicheng Hongbao Veterinary Medicine Co., Ltd., Jiangxi, China) and Yunnan Baiyao (Yunnan Baiyao Group Co., Ltd., Yunnan, China) were applied to the wound after the sampling.

### 2.2. RT-qPCR Analysis

Total RNA was extracted from the skin samples of the Gansu Alpine Merino using Trizol Reagent (Shanghai Yuanye Biotechnology Co., Ltd., Shanghai, China) as per the manufacturer’s instructions. An ultraviolet spectrophotometer was used to determine the purity and concentration of the extracted RNA, which was subsequently stored at −80 °C. The Prime Script^TM^ RT reagent kit (Nanjing Novizan Biotechnology Co., Ltd., Nanjing, China) was used for RNA reverse transcription as per the manufacturer’s instructions. The cDNA was stored at −20 °C until further analysis.

*β-actin* was used as the internal reference gene. Primers were constructed using the NCBI database and Primer Premier 5 software (Premier Biosoft, Palo Alto, CA, USA). The primer sequences and PCR conditions are given in Table 1. The qPCR was performed using a Biosystems QuantStudio^R^ 6 Flex (Thermo Lifetech, Waltham, MA, USA) according to the manufacturer’s instructions. The SYBR Green Pro Taq HS qPCR Kit (Accurate Biology, Hunan, China) was used to perform the RT-qPCR with cDNA as a template. The thermal cycle parameters were as follows: initial denaturation at 95 °C for 10 min, followed by 45 denaturation cycles of 15 s at 95 °C and 1 min of annealing, and extension at 60 °C. At each time of sample collection (1, 30, 60, and 90 days), there were 3 biological replicates and 4 technical replicates to ensure that the findings were genuine and trustworthy.

### 2.3. Immunohistochemical Analysis

Paraffin sections were typically deparaffinized using gradient alcohol and distilled water, and washed three times with PBS (each time for 5 min). Citric acid antigen retrieval buffer (pH = 6.0) was used for antigen retrieval; the sections were washed three times with PBS, (each time for 5 min), incubated with 3% hydrogen peroxide solution for 25 min at room temperature in the dark, rinsed three times with PBS (5 min each time), and incubated with 3% bovine serum albumin (BSA) for 30 min at room temperature. PBS was produced in proportion to the primary antibody (*Dlx3* rabbit anti-human-mouse polyclonal antibody Abmart/PA39005 and *Lef1* rabbit anti-human-mouse monoclonal antibody Abmart/T553505), and the slices were treated overnight in the box at 4 °C. Using a PBS shaker, the sections were washed thrice for 5 min each. After drying the slices, the corresponding secondary antibody was applied dropwise and incubated at room temperature for 50 min. The slices were dried and mixed with a freshly prepared diaminobenzidine chromogenic solution until a positive (brown-yellow) color development was observed. The sections were placed under running water to terminate the color development. They were subjected to hematoxylin staining for 3 min until the color turned to blue; further washed with running water, dehydrated, and turned transparent; and mounted with neutral gum. The sections were examined under a microscope, and the images were collected and analyzed.

### 2.4. Immunofluorescence Analysis

Paraffin sections were conventionally dewaxed using gradient alcohol and distilled water. Antigen retrieval was performed using ethylene diamine tetra-acetic acid antigen retrieval buffer (pH = 8.0) for antigen repair. It was ensured that the buffer was excessively evaporated throughout this procedure, resulting in dry slices. They were allowed to cool naturally before washing three times using a PBS shaker for 5 min each and spinning dry PBS. BSA was added dropwise to block the sections for 30 min. The experiment was performed using the same procedure as that for the immunohistochemistry (addition of primary and secondary antibodies with PBS washing in-between). After gentle shaking and drying the slices, DAPI dihydrochloride (DAPI) dye was added dropwise to the circle and incubated at room temperature for 10 min in the dark. The slices were washed three times with PBS for 5 min each. An autofluorescence quencher was added to the circle for 5 min and washed with running water for 10 min. Anti-fluorescence quenching mounting tablets were used to mount the sections after they were gently dried. The sections were examined under a microscope, and the images were collected and analyzed.

### 2.5. Measurements and Statistical Analysis

HE-stained sections were scanned using a PANNORAMIC (3DHISTECH Ltd., Budapest, Hungary) panoramic slice scanner, and the images were analyzed using Image-ProPlus6.0 software (Media Cybernetics Inc., Rockville, MD, USA) to calculate the number of hair follicles, hair follicle area, number of hair follicles per unit area, epidermal thickness, dermal thickness, and hair follicle diameter. The PANNORAMIC panoramic slice scanner was used for scanning and imaging for immunohistochemistry and immunofluorescence, and the pictures were processed using Image-ProPlus6.0 software (Media Cybernetics Inc., Rockville, MD, USA). For each slice, three fields of view were selected to calculate the average optical density of the positive reactant and distribution density of various regions. To calculate the RT-qPCR results, the −2^ΔΔCT^ method [24] was used in Excel software. SPSS 22.0 (SPSS Inc., Chicago, IL, USA) statistical software was used to perform a one-way analysis of variance on all the above data. The results were expressed as mean ± standard error (S.E.) deviation; *p* < 0.05 was considered significant.

## 3. Results

### 3.1. Expression of Lef1 and Dlx3 in the Skin Tissues of Gansu Alpine Merino at Various Ages

The RT-qPCR results revealed that both *Lef1* and *Dlx3* were expressed during hair follicle development. The mRNA levels differed significantly according to the age of the lambs, with the mRNA level in the skin at 30th day being significantly higher than that of the other three age groups (*p* < 0.05; Figure 1). The trends in the mRNA levels of *Lef1* and *Dlx3* remained consistent, both exhibiting an increase followed by a decrease.

### 3.2. Histomorphological Analysis and Evaluation of Hair Follicles at Various Ages

HE staining was performed to analyze the skin development of Gansu Alpine Merino at various ages (Figure 2), including the number of hair follicles per unit area, number of hair follicles, area of hair follicles, thickness of the epidermis, thickness of the dermis, and diameter of hair follicles. The number of hair follicles per unit area at 1 day was significantly higher than that at 60 and 90 days (*p* < 0.05; Figure 3A). The number of hair follicles at 1 day was significantly higher than that at 60 and 90 days (*p* < 0.05; Figure 3B). The area of hair follicles at 1 day was significantly lower than that at 90 days (*p* < 0.05; Figure 3C). The thickness of the dermis was significantly higher at 90 days than at 1 day (*p* < 0.05; Figure 3E). The number of hair follicles per unit area and total number of hair follicles of Gansu Alpine Merino were the highest at 1 day and the lowest at 90 days, and they decreased with the increasing age. The area of hair follicles and hair follicle diameter were the smallest at 1 day and largest at 90 days, indicating that they increased with age. Epidermal thickness and hair follicle diameter at various ages exhibited little change, and no significant difference was observed among the four groups (*p* > 0.05; Figure 2F and Figure 3D).

### 3.3. The Average Optical Density of Lef1 and Dlx3 in the Skin Tissues of Gansu Alpine Merino at Various Ages

The expression of Lef1 and Dlx3 in the skin tissues of Gansu Alpine Merino at various ages was analyzed using immunohistochemical staining combined with average optical density statistics (Figure 4). The average optical density results revealed that both Lef1 and Dlx3 proteins were expressed during hair follicle development. The protein levels varied significantly at different periods. The protein expression in skin at 1-day-old lambs was significantly lower than that in the other three groups (*p* < 0.05; Figure 5A,B), and it reached a peak at 60 days. The protein expression trends of Lef1 and Dlx3 remained consistent, both exhibiting an increase followed by a decrease.

### 3.4. Distribution of Lef1 and Dlx3 in the Skin Tissues of Gansu Alpine Merino at Various Ages

The immunofluorescence technique was used to analyze the distribution of Lef1 and Dlx3 in the skin tissues of Gansu Alpine Merino at various ages (Figure 6 and Figure 7). The results revealed that Lef1 in the skin tissue of 1-day-old Gansu Alpine Merino exhibited a high density and strong positive expression in the outer root sheath, hair medulla, sebaceous glands, and dermal papilla (Table 2). Strong positive expression was observed in the stratum corneum and inner root sheath. Moreover, 30-, 60-, and 90-day-old skin tissues only exhibited a high density and strong positive expression in the dermal papilla.

The results revealed that Dlx3 was expressed in the stratum corneum, inner root sheath, outer root sheath, hair medulla, sebaceous glands, and dermal papilla in skin tissues of 1- and 90-day-old Gansu Alpine Merino with a high density and strong positive expression (Table 2). The 30-day-old skin tissues exhibited a high density and strong positive expression in inner root sheath, outer root sheath, and dermal papilla; strong positive expression in the stratum corneum and sebaceous glands; and moderate positive expression in the hair medulla. The 60-day-old skin tissues exhibited a high density and strong positive expression in the stratum corneum, inner root sheath, outer root sheath, sebaceous glands, and dermal papilla; and strong positive expression in the hair medulla.

## 4. Discussion

The growth and development of hair follicles is a complex and long-term physiological process that is regulated by various physical factors and signal pathways. Gene expression is one of the decisive factors that affects hair growth in sheep [1,2]. Among multiple signaling pathways, the Wnt/β-catenin signaling pathway has a considerable influence on the embryogenesis of hair follicles and hair growth [3,4]. Therefore, the Wnt/β-catenin signaling pathway contributes to the development and maturation process of hair follicles.

*Lef1* can regulate the Wnt/β-catenin signaling pathway, and acts as one of the main members downstream of the pathway [5]. Further, *Dlx3* can intervene in the development of hair follicles by regulating the proliferation of dermal papilla [12]. Exploring the expression pattern of *lef1* and *Dlx3* during hair follicle development and maturation will lay the foundation for understanding the mechanism of hair follicle maturation. In this study, the expression, location, and biological functions of *Lef1* and *Dlx3* in the lateral skin of Gansu Alpine Merino were analyzed systematically. 

### 4.1. Study on the Morphological Development of Skin and Hair Follicles at Various Ages

There are two types of hair follicles: primary and secondary. They distribute in groups in the skin to form follicular clusters. In most breeds of sheep, the hair follicle complex is a cluster of three primary follicles and many secondary follicles, which are surrounded by sebaceous glands [25]. Skin development, folliculogenesis, histomorphology, and spatiotemporal expression of genes in various periods are mostly reported in humans [16], mice [20,21], and other study subjects in the embryonic stage. Studies on the development of hair follicles at various growth stages after birth are scarce, and those on the development of hair follicles at various growth stages after birth in fine-wool sheep are not available. Sheep is born with fully developed primary hair follicles. Further, the number of hair follicles per unit area gradually decreases with age, and secondary hair follicles gradually mature and increase with age. The occurrence of both physiological phenomena is concentrated in the first 3 months after birth, and then it tends to stabilize [25]. Therefore, in this study, we selected Gansu Alpine Merino as the representative of fine wool sheep and collected skin samples at 1, 30, 60, and 90 days to study the histomorphology and hair follicle development. 

HE staining revealed that the numbers of hair follicles and hair follicles per unit area in 1-day-old Gansu Alpine Merino were the highest among the four ages. The dermis exhibited a certain degree of thickening with the increase in age, which indicated that the number of hair follicles and thickness of the dermis were not constant after birth, but exhibited certain changes with the increasing age. These results were consistent with those of a related study reporting that primary hair follicles were completely developed after birth in Gansu Alpine Merino, and primary follicular density was the highest at 1 month of age and decreased with the growth of lambs [26,27], including reformation of substrate and dermal condensates and thickening of the epidermis and dermis. Moreover, this change was studied in chicken and geese [28], as well as mice [29].

### 4.2. mRNA and Protein Expression Studies of Lef1 and Dlx3 in Skin Tissues at Various Ages

In this experiment, the mRNA levels of *Lef1* and *Dlx3* in the skin of Gansu Alpine Merino at various developmental stages were examined using RT-qPCR. The results revealed that both *Lef1* and *Dlx3* were expressed during hair follicle development. The mRNA levels differed significantly in different periods, with the mRNA in 30-day-old skin being significantly higher than that in the other three groups. The average optical density results revealed that both Lef1 and Dlx3 proteins were expressed during hair follicle development, and the protein expression levels differed significantly at different ages; the protein expression in 1-day-old skin was significantly lower than that in the other three groups, which reached a peak at 60 days. However, the mRNA levels of *Lef1* and *Dlx3* were not exactly consistent with the protein expression levels, which may have been due to the differences in peak expression of mRNA and protein at the transcriptional level [30].

The trends of *Lef1* and *Dlx3* mRNA and protein expression remained consistent, both exhibiting an increase followed by a decrease. Therefore, combined with the studies on the maturation of hair follicles in Gansu Alpine Merino (i.e., the primary hair follicles were developed during the embryonic period, whereas the secondary hair follicles began to mature only in the late embryonic period), the maturation rate of secondary hair follicles was 49.59% at birth and 92.40% at 3 months of age. Particularly, within 2 months of age after birth, the fastest maturation observed was 91.61% at 2 months of age [25]. In contrast, in this study, we observed that the peak mRNA and protein levels of *Lef1* and *Dlx3* occurred at the 30th and 60th days, respectively; however, both peaks occurred during the period of fastest maturation of the secondary hair follicles. Furthermore, in a study of spatiotemporal expression patterns, it was observed that *Lef1* was an important regulatory gene in the Wnt pathway for the induction of primary feather follicles and a positive regulator of primary feather follicle development in goose skin [31], while *Lef1* also had an effect on the growth and development of swan and gosling hair follicles [32]. In contrast, a weighted gene co-expression network analysis (WGCNA) was applied in velvet goats to identify *Lef1* as a key factor in hair follicle cycle development, and it was further determined to play an important role in hair follicle cycle development using RT-qPCR [33]. It was also found that *Dlx3* was screened as a key transcription factor in the hair follicle cycle in yak based on a WGCNA analysis, and further identified by immunofluorescence staining and Western blotting to play an important role in hair follicle cycle [34]. Our results suggested that *Lef1* is a positive regulator of secondary hair follicle maturation in Gansu Alpine Merino, and *Dlx3* is a downstream factor of *Lef1* and is positively regulated by *Lef1*. Therefore, *Dlx3* may also be a positive regulator of secondary hair follicle maturation.

### 4.3. Localization Study of Lef1 and Dlx3 in Skin Tissues at Various Ages

Immunofluorescence results revealed that Lef1 was mainly expressed in the dermal papilla in the skin of Gansu Alpine Merino after birth, with a high density and strong positive expression. Dlx3 was mainly expressed in the inner root sheath, outer root sheath, and dermal papilla, with a high density and strong positive expression. This suggested that Lef1 acts mainly on the hair papilla to regulate the maturation of secondary hair follicles, whereas Dlx3 acts mainly on the hair papilla, inner root sheath, and outer root sheath to participate in the maturation process of secondary hair follicles. These findings were consistent with some previous studies reporting that Lef1 was expressed in the development of the hair bulb closer to the dermal papilla [35], and Dlx3 was widely expressed in the hair shaft, hair matrix, and inner root sheath [36]. 

In addition, *Lef1* was a central effector that mediated the classical pathway of Wnt signaling, and was involved in feather follicle morphogenesis on dermal papillae [30]. In contrast, selective elimination of *Dlx3* led to complete hair loss due to unformed hair shaft and inner root sheath [17,37]. Dermal papilla cells are located at the base of the hair follicles and secrete various cytokines that regulate neighboring tissues, thereby regulating hair growth and renewal [38,39]. Therefore, the dermal papilla is a very important organ during the maturation, development, and growth of hair follicles. Additionally, this may be related to the fact that hair cells can receive and send signals [40]. Overexpression of *Dlx3* was reported to promote the proliferation of dermal papillae in velvet goat dermal papillae cells, whereas downregulation of *Dlx3* decreased the proliferation of dermal papillae [41]. This indicated that *Lef1* and *Dlx3* act first on the hair papilla cells during secondary hair follicle maturation to promote secondary hair follicle maturation, and later, as the hair follicle matures, *Dlx3* expresses in the inner and outer root sheath to promote normal maturation of secondary hair follicles, thus ensuring normal hair growth.

### 4.4. Prospects for Clinical Applications of Lef1 and Dlx3

Ectodermal dysplasias (ED) is a family of genetic skin diseases in which the common clinical sign is hypertrichosis, and inactivation of *Lef1* targeting in mice leads to complete blockage of ectodermal development [42]. A new study confirmed that single allele and double allele variants in *Lef1* are the etiology of a new syndrome leading to ED [43]. Hutchinson–Gilford progeria syndrome (HGPS) is a deleterious premature aging disease with common clinical symptoms of alopecia and skin sclerosis. It was found that *Lef1* expression was reduced in the early stage of the keratin-forming cell differentiation of HGPS-derived induced pluripotent stem cells (iPSCs), which also suggests an important role of *Lef1* in the formation of HGPS [44]. There is a transient regenerative cell type in the developing skin known as papillary fibroblasts, which are the primary source of new hair papillae during skin development [45], and are regulated by the expression of the typical Wnt transcription factor *Lef1*, which in turn was identified as a regenerative factor in the skin that is used for skin and hair follicle regeneration [46]. It was found that *Dlx3* had an effect on hair follicle development via RNA sequencing (RNA-seq) of velvet goat skin tissue [47], while *Dlx3* was also screened using WGCNA and found to play an important role as a key transcription factor in the hair follicle cycle in yaks [34]. It also was shown that miR-22 regulated hair follicle cyclic changes and affected the formation of IRS and HS by repressing the expression of transcription factor *Dlx3* [48]. Genetic exploration of sheep and goats has revealed a number of genes associated with hair follicle development that may have an impact on fiber characteristics [49]. However, the research and application of *Lef1* and *Dlx3* in hair follicles and hairs are still mostly in the candidate stage, and further clinical studies are needed, so this should be the direction of future research.

## 5. Conclusions

The results of this study indicated that: (1) in the rapid maturation stage of secondary hair follicles, the expression of *Lef1* and *Dlx3* significantly increases, which promotes the maturation of secondary hair follicles; (2) *Lef1* is mainly expressed in the dermal papilla and plays an important role in the maturation of secondary hair follicles; and (3) *Dlx3* is mainly expressed in the dermal papilla and inner and outer root sheaths, and plays an important role in the maturation of secondary hair follicles. Further studies are needed to elucidate the processes underlying this regulatory mechanism.

## Figures and Tables

**Figure 1 genes-13-01326-f001:**
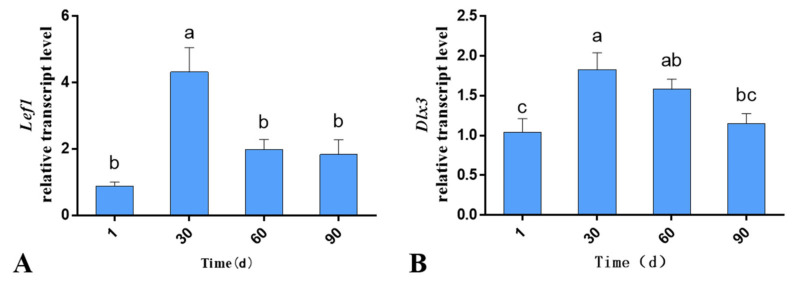
mRNA levels of skin-related genes in Gansu Alpine Merino at various ages. The mRNA levels of *Lef1* and *Dlx3* were normalized to the level of *β-actin* mRNA. (**A**) Relative mRNA levels of *Lef1* at various ages; (**B**) relative mRNA level of *Dlx3* at various ages. The data are expressed as the mean ± S.E. Different letters indicate significant differences between different ages (*p* < 0.05).

**Figure 2 genes-13-01326-f002:**
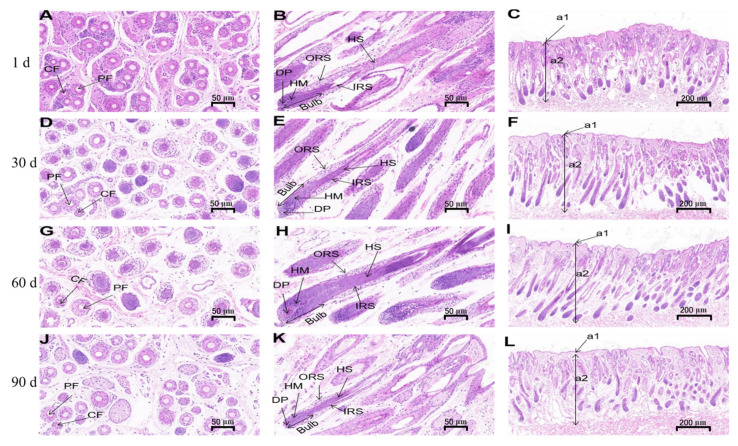
HE staining of skins of Gansu Alpine Merino at various ages. (**A**,**D**,**G**,**J**) Longitudinal section (20×); PF: primary hair follicle and CF: secondary hair follicle. (**B**,**E**,**H**,**K**) Transverse section (20×); HS: hair shaft, ORS: outer root sheath, IRS: inner root sheath, Bulb: hair bulb, DP: dermal papilla, and HM: hair matrix. (**C**,**F**,**I**,**L**) Transverse section (6.6×); a1: epidermis and a2: dermis.

**Figure 3 genes-13-01326-f003:**
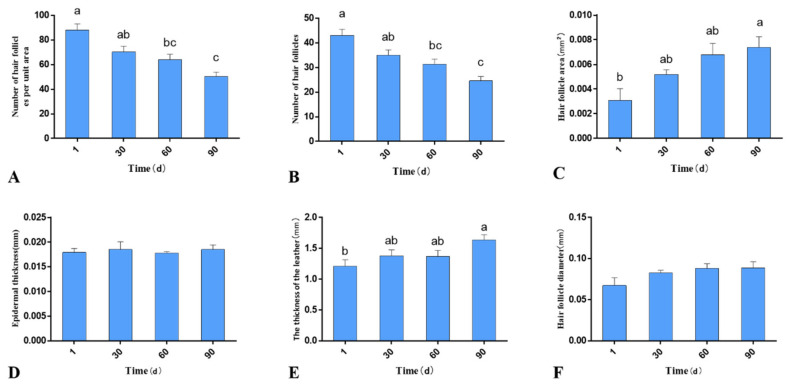
HE staining analysis of skins of Gansu Alpine Merino at various ages. (**A**) The number of hair follicles per unit area at various ages; (**B**) the number of hair follicles at various ages; (**C**) hair follicle area at various ages; (**D**) thickness of the epidermis at various ages; (**E**) thickness of the dermis at various ages; (**F**) diameter of hair follicles at various ages. The data are expressed as the mean ± S.E. Different letters indicate significant differences between different ages (*p* < 0.05).

**Figure 4 genes-13-01326-f004:**
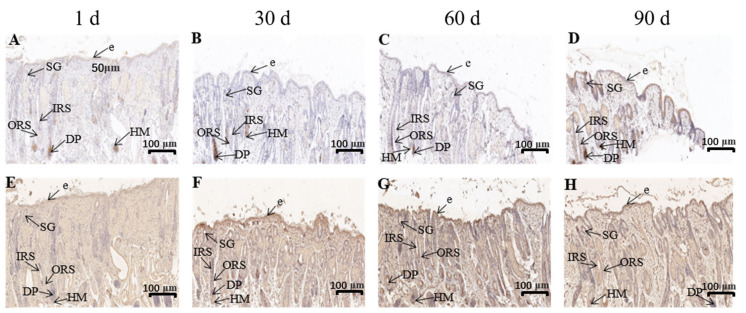
Immunohistochemical staining (10×) of skin tissues of Gansu Alpine Merino at various ages. (**A**–**D**) Lef1 skin tissue distribution at various ages; (**E**–**H**) Dlx3 skin tissue distribution at various ages. ORS: outer root sheath; IRS: inner root sheath; DP: dermal papilla; HM: hair matrix; e: epidermis; SG: sebaceous glands.

**Figure 5 genes-13-01326-f005:**
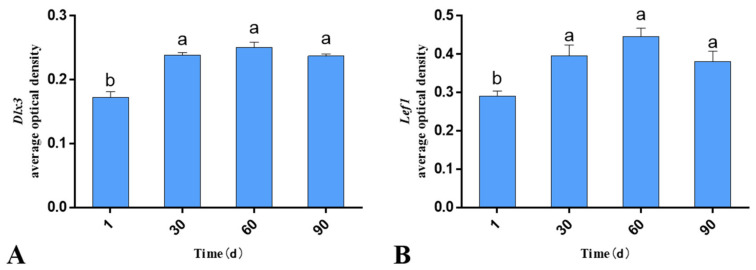
Statistics of the average optical density of the skin tissue of Gansu Alpine Merino at various ages as per immunohistochemical analysis. (**A**) Average optical density of Dlx3 at various ages; (**B**) average optical density of Lef1 at various ages. The data are expressed as the mean ± S.E. Different letters indicate significant differences between different ages (*p* < 0.05).

**Figure 6 genes-13-01326-f006:**
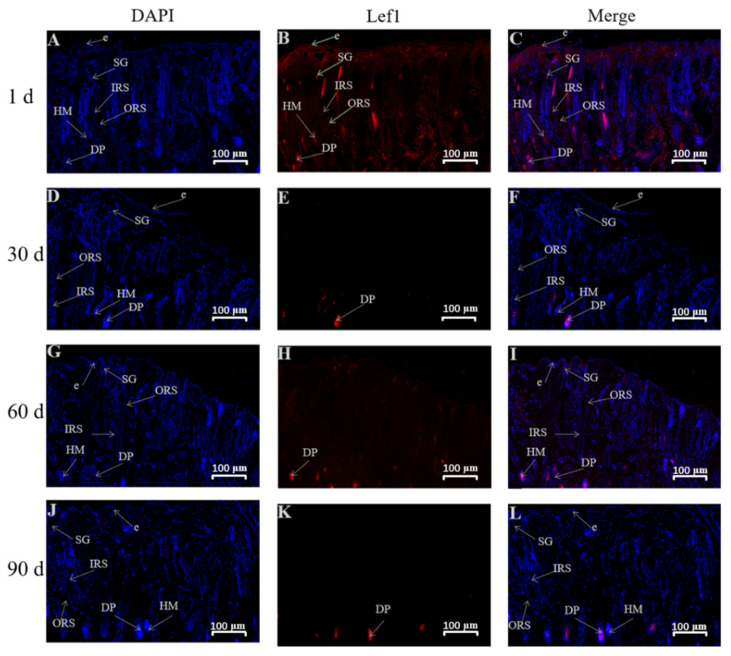
Immunofluorescence staining for Lef1 in skin tissues of Gansu Alpine Merino at various ages (10×). (**A**,**D**,**G**,**J**) The blue-colored tissue in the figure shows the DAPI-labeled nuclear fluorescence coloring; (**B**,**E**,**H**,**K**) the red-colored tissue shows the fluorescence coloring of Lef1; (**C**,**F**,**I**,**L**) Fluorescent coloring after merging Lef1 with DAPI. ORS: outer root sheath; IRS: inner root sheath; DP: dermal papilla; HM: hair matrix; e: epidermis; SG: sebaceous glands.

**Figure 7 genes-13-01326-f007:**
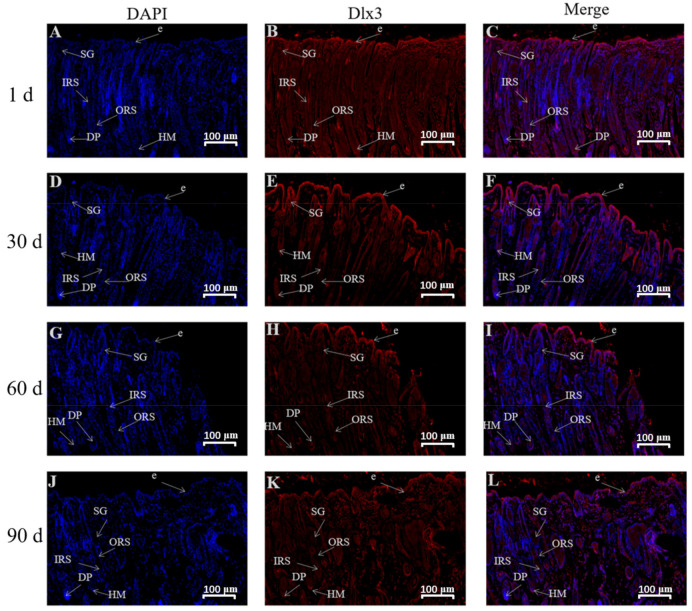
Immunofluorescence staining for Dlx3 in skin tissues of Gansu Alpine Merino at various ages (10×). (**A**,**D**,**G**,**J**) The blue-colored tissue shows the DAPI-labeled nuclear fluorescent coloring; (**B**,**E**,**H**,**K**) the red tissue shows the fluorescent coloring of Dlx3; (**C**,**F**,**I**,**L**) Fluorescent coloring after merging Dlx3 with DAPI. ORS: outer root sheath; IRS: inner root sheath; DP: dermal papilla; HM: hair matrix; e: epidermis; SG: sebaceous glands.

**Table 1 genes-13-01326-t001:** Primer sequences and annealing temperatures used for RT-qPCR.

Gene	GenBank Accession No.	Primer Sequence (5′-3′)	Product Size (bp)	Annealing Temperature (°C)
*Lef1*	XM_042251154.1	F: ACCATGACAAGGCCAGAGAA	228	60
		R: TGATGAGAGGGGTGAGAGGA		
*Dlx3*	XM_004012779.5	F: AACCGCCGTTCCAAATTCAA	230	60
		R: TGTGCATGGTACCAGGAGTT		
*β-actin*	NM_001009784	F: AGCCTTCCTTCCTGGGCATGGA	113	60
		R: GGACAGCACCGTGTTGGCGTAGA		

**Table 2 genes-13-01326-t002:** Distribution density of Lef1 and Dlx3 in various parts of skin tissues at various ages.

Names	Days	Corneum	Inner Root Sheath	Outer Root Sheath	Hair Medulla	Sebaceous Gland	Dermal Papilla
Lef1	1	+++	+++	++++	++++	++++	++++
	30	−	−	−	−	−	++++
	60	−	−	−	−	−	++++
	90	−	−	−	−	−	++++
Dlx3	1	++++	++++	++++	++++	++++	++++
	30	+++	++++	++++	++	+++	++++
	60	++++	++++	++++	+++	++++	++++
	90	++++	++++	++++	++++	++++	++++

−, No positive expression; ++, medium positive expression; +++, strong positive expression; ++++, high-density strong positive expression.

## Data Availability

The authors affirm that all data necessary for confirming the conclusions of the article are present within the article, figures, and tables.

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
