# Peer review of "Lef1 and Dlx3 May Facilitate the Maturation of Secondary Hair Follicles in the Skin of Gansu Alpine Merino"

_genes, 2022, doi:10.3390/genes13081326_

Round 1

Reviewer 1 Report

Introduction is OK

M & M

Table 1. There is a small typo in one of the primers, please check. Also, please include all the details of the PCR, e.g., product size etc. The corrected table must be transferred to supplementary material.

2.3. and 2.4. are standard pathology techniques, no need for full description. Please delete.

Results

Figures 1,3, 5. Please colourize the bars.

Discussion

This is badly organised. Please divide in sub-sections and please use shorter paragraphs. Also, please add a brief passage with clinical applications of the findings. Finally, some recent significant references are missing and should be added.

Overall. The manuscript must be corrected as indicated above and then re-evaluated.

Reviewer 2 Report

Review:

The authors focused on the transcription factors (Lymphatic enhancer factor 1 (Lef1) and distal-less homeobox 3 (Dlx3)), which are responsible for the regulation of hair sleeves in Gansu Alpine Merino sheep.

The rate of wool growth can vary over a wide range due to genotype and the influence of various physiological and environmental factors. The people, but the precise mechanism is still being discussed.

The maximum rate at which an animal can produce wool or hair and the range of variation possible in several characters related to quality, are set by its genotyp. There are definite differences between breasts in capacity to grow, wool and in various fleece characteristics. Thus, Merinos, which have a much greater follicle density than down and long wool breeds,

a simultaneous mass of wool, but more than the breeds down. However, there is a significant variation in the rate of wool growth between strains and individual sheep.

The maximum rate at which an animal can produce wool or hair and the range of variation possible in several characters related to quality, are set by its genotyp. There are definite differences between breasts in capacity to grow, wool and in various fleece characteristics. Thus, Merinos, which have a much greater follicle density than down and long wool breeds, a similar mass of wool but more than the down breeds. However, there is a significant variation in the rate of wool growth between strains and individual sheep. In the Australian Merino, a comparison between the fine, medium and strong wool strains shows an increasing clean fleece weight associated with increased fibre diameter, staple length and body weight. Genetically high-producing Merino sheep have follicles which are remally straighter and deeper in the skin, wachst einen niederen Sulphur content and have a lower concentration of cystine in plasma than Low-Producing sheep; the number of follicles pro unit area of skin is sometimes. The lower sulphur content of the wool is due to a lower content of ultra high-sulphur proteins. If increased wool production is achieved in a breeding program without increasing the number of follicles, the rate of fiber production must be improved by individual follicles. An inevitable result is an increase in mean diameter or length growth rate of fibres or both. The characteristics of fibers and folk are particularly heritable and significant changes can be determined by selecting the characteristic characteristics. The heritability of wool traits such as greasy or clean wool weight, number of follicles per unit area of skin, S/P follicle ratio, fibre diameter, staple length and crimp frequency are in the region of 0. 3 to 0. 6. Diese Traits sind korrelated and account must be taken of this, if it is desired to increase wool weight without altering various characteristics of the fleece.

 Methodology:

In addition to the determination of molecular transcription factors Lymphatic enhancer factor 1 (Lef1) and distal-less homeobox 3 (Dlx3)

In addition to the molecular determinations, it is worthwhile to analyse the composition of the wool on the composition and content of the amino acids methionine, serine, cystine, cysteine, lysine and also microelements, especially zinc in the blood.

The analytical methods used are unquestionable

Please explain why only 3 individuals were used in the RT-qPCR study? Similarly, 3 individuals were used for immunohistological and ummunofluorescence analyses. Does this small number provide statistical certainty?

Variations in the supply of nutrients to follicles can have a significant influence on the rate of fiber production and the characteristics of the fleece. Most sheep and goats are kept under free-ranging conditions and the quantity of feed available to them may vary throughout the year. Consequently, the peak rate of wool growth is frequently two to three times the minimum rate for the spread of sheep. Controlled feed experiments have achieved the large effect of feed-intake on the rate of wool growth; triple to fourfold changes in the rate of wool growth may be induced in a single animal. There is less experimental evidence available to regulate the effects of diet on the fiber production of goats. There is no general agreement on the precise form of the relationship between the will to promote growth and feed intake. Available evidence points to a positive linear relationship between intake of digestible try matter and wool growth, and states there is no unequivocal evidence for a curvilinear relationship although, this may occur at rates of wool growth approaching the genetic potential. It has been suggested that wool growth rate is influenced by the extent and direction of body weight change.

 It is regrettable that the second breed of the experiment was designed to obtain the results of Lef1 and Dlx3 of the tested sheep of the breed Gansu Alpine Merino.

Results:

The results are presented in the form of 7 drawings and 2 drawings.

RT-qPCR results showed that both Lef1 and Dlx3 were expressed during hair follicle development. MRNA levels were significantly different depending on the age of the animals, with skin mRNA levels significantly higher at day 30 than in the other three age groups. The tissue was stained in the collected skin samples to analyse the development of Gansu Alpine Merino skin. It is regrettable that the histological studies were not compared with the blood metabolites, in particular with the sulphurous amino acids methionine, serine, cystine, cysteine, lysine and also with the markings in the wool of trace elements including zinc and copper.

Discussion:

The discussion chapter is well written, although the scope of data including analysis of gene expression of 2 genes and histochemical studies is very limited. In future, physiological as well as ecological and nutritional factors should be taken into account.

General remark

 I think the article should be corrected before publication.

Round 2

Reviewer 1 Report

The manuscript has been improved and is almost ready for acceptance. However, the authors must perform a careful reading, in order to correct various linguistic problems scattered throughout the manuscript. Then the manuscript will be acceptable.